# A Nonlinear-Model-Based High-Bandwidth Current Sensor Design for Switching Current Measurement of Wide Bandgap Devices

**DOI:** 10.3390/s23104626

**Published:** 2023-05-10

**Authors:** Xia Du, Liyang Du, Yuxiang Chen, Yuqi Wei, Andrea Stratta, Homer Alan Mantooth

**Affiliations:** Department of Electrical Engineering, University of Arkansas, Fayetteville, AR 72701, USA; liyangdu@uark.edu (L.D.); yc041@uark.edu (Y.C.); yuqiwei@uark.edu (Y.W.); astratta@uark.edu (A.S.); mantooth@uark.edu (H.A.M.)

**Keywords:** current sensor, current transformer, high bandwidth, low cost, power electronics applications, wide bandgap devices, silicon carbide devices, switching current measurement, nonlinear modeling

## Abstract

With the growing adoption of wide bandgap devices in power electronic applications, current sensor design for switching current measurement has become more important. The demands for high accuracy, high bandwidth, low cost, compact size, and galvanic isolation pose significant design challenges. The conventional modeling approach for bandwidth analysis of current transformer sensors assumes that the magnetizing inductance remains constant, which does not always hold true in high-frequency operations. This can result in inaccurate bandwidth estimation and affect the overall performance of the current sensor. To address this limitation, this paper provides a comprehensive analysis of nonlinear modeling and bandwidth, considering the varying magnetizing inductance in a wide frequency range. A precise and straightforward arctangent-based fitting algorithm was proposed to accurately emulate the nonlinear feature, and the fitting results were compared with the magnetic core’s datasheet to confirm its accuracy. This approach contributes to more accurate bandwidth prediction in field applications. In addition, the droop phenomenon of the current transformer and saturation effects are analyzed in detail. For high-voltage applications, different insulation methods are compared and an optimized insulation process is proposed. Finally, the design process is experimentally validated. The bandwidth of the proposed current transformer is around 100 MHz and the cost is around $20, making it a low-cost and high-bandwidth solution for switching current measurements in power electronic applications.

## 1. Introduction

Power electronics applications, which use power semiconductor-based topology and driving techniques, offer more flexible and efficient solutions for power conversion. In recent years, wide bandgap (WBG) power devices, such as silicon carbide (SiC) and gallium nitride (GaN), have gained increasing attention and adoption in various fields, including automotive, renewable energy, and consumer electronics [1,2,3,4,5] to provide higher efficiency, switching frequency, and power density [6,7,8]. To research the properties and behaviors of WBG devices, the switching current measurement becomes an increasingly important study approach [9,10,11,12,13,14,15]. Moreover, overcurrent protection is an essential functionality that prevents semiconductor switches from failure, especially in high-power applications at medium voltage levels, where the components are expensive and a failure may lead to catastrophic consequences [16,17,18]. Furthermore, it is also necessary to feedback the switching current to the driving system to realize current balancing for WBG paralleling operations [19,20,21,22,23]. However, the much higher switching frequency and faster switching transient brings more challenges to measure the switch current accurately [24,25]. Therefore, designing current sensors for switching current measurement and feedback control is critical to accelerate the development and implementation of new devices and techniques in power electronics.

With the trend toward high-frequency operation in the megahertz (MHz) range and high-power-density concepts in power conversion systems, current sensor design for switching current measurement has become more challenging and needs to meet much higher requirements [26,27]:(1)Bandwidth higher than 50 MHz: WBG devices have a faster switching speed compared with traditional silicon-based devices, which leads to faster rising and falling edges of the current waveform during switching transients. To accurately measure these fast-changing currents, a high-bandwidth current sensor is required. To illustrate this point, the SiC MOSFET SCT3120AL is tested, and the switching transient current is measured as shown in Figure 1. The bandwidth of a waveform is mainly determined by the slew rate as well as any oscillations or resonances that may be present in the signal. To capture the fast-rising time *t_r_* 18.8 ns, the minimum required bandwidth can be estimated according to (1) [9,10,28,29,30,31], which is 18.6 MHz, and in practice, it is recommended to have a bandwidth that is three to five times higher than the highest frequency [28,29,30,31] in order to ensure sufficient accuracy and avoid signal distortion, which is 55.85 MHz to 93 MHz. The current oscillation is caused by the output capacitance *C_oss_* of SiC devices and power loop inductance *L_loop_*. The oscillating frequency *f_o_* can be calculated by (2) [31,32]. Typically, *C_oss_* ranges from 50 pF to 200 pF and *L_loop_* is around 100 nH, thus leading to a minimum oscillation frequency of 10 MHz. Therefore, to measure the switching current accurately, the required bandwidth should be above 50 MHz.
(1)$$BW=\frac{0.35}{t_{r}}$$
(2)$$f_{o}=\frac{1}{2\pi \sqrt{C_{oss}\cdot L_{loop}}}$$

(2)Small size suitable for TO-237 package of discrete devices or integration within the power modules: the preference for a smaller size in current sensor design is driven not only by the trend towards high power density in power electronics applications, but also by the growing focus on integration within the power modules or with discrete devices with the TO-237 package [31,33,34,35]. Additionally, in high-frequency operation, the parasitic effects can become significant and cause unintended oscillations and voltage spikes, which can potentially damage the power devices and degrade the overall system performance [35,36]. A smaller current sensor can reduce the effects of parasitic inductance and capacitance in the circuit.(3)Galvanic insulation capability up to 10 kV: with the trend towards a higher operating voltage 10~15 kV for WBG devices [37,38,39], the electrical stress on the power devices and components in the system also increases, which leads to a higher risk of insulation degradation and breakdown. Therefore, galvanic insulation is necessary to provide a barrier between the power side and controller side. This helps to protect against electrical faults and improve the overall safety and reliability of the system.(4)Low cost: for high-power, high-voltage applications in power electronics, a series of converter topologies, including modular multilevel converters (MMCs), cascaded H-bridge converters (CHBs), and neutral-point-clamped (NPC) converters [40,41,42], usually contain more than 50 switch units. To measure the switching current, the same number of current transformers is required, bringing a significant extra cost for the system. Therefore, devising a low-cost current measurement solution is necessary to make the switching current measurement economically feasible.

The existing commercial current sensors for switching current measurement of WBG devices include Hall effect current sensors, coaxial current shunt, Rogowski coil, and current transformers. However, most of them cannot meet most of the requirements mentioned above. Among these candidates, current transformers exhibit promising features compared with other sensors. They have higher bandwidth (up to 250 MHz) [43] than Hall effect current sensors (up to 1 MHz) [44] and they are non-contact and non-invasive compared with coaxial current shunt [45,46]. They also have a lower cost and simpler construction than Rogowski coils [47]. However, current transformers also have limitations for switching current measurement such as nonlinearity and saturation effects at high current levels and large size. Therefore, it is necessary to further research the characteristics and design method of current transformers to improve the performance for power electronics applications.

Several efforts have been dedicated to current transformer design to improve density, accuracy, and bandwidth. In [48], a current transformer was proposed with a small sized outer diameter of 10 mm, inner diameter 6 mm, and height 4 mm. It was designed for overcurrent protection and tested under a 30 A double pulse test. However, the performance under continuous current pulse was not covered, and the design process was not mentioned. In [13,31,49], a cascade connection structure was applied on the current sensor to reduce the size. The size of the first-stage current transformer was sufficiently small, while the output signal of the first stage was weak, and a large-scale and high-accuracy current sensor was required in the second stage, which added to the complexity, size, and cost of the entire measurement system. In [50], a general high-frequency modeling approach was built, and the impacts of several components on bandwidth were analyzed, but the bandwidth performance is not quantified and experimentally verified. In addition, the aforementioned literature does not include the analysis of the nonlinearity of the magnetic core. Motivated by the incomprehensive analysis and design of current transformer sensors, this paper demonstrates nonlinear modeling for magnetizing inductance across a wide frequency range, analyzes the droop and saturation issues in detail, and provides a comprehensive guide for high-accuracy current transformer design for switching current measurements. To cater to high-voltage power electronics applications, the insulation process is optimized. Finally, all of the design and analyses are validated in a continuous 10 μs switching current pulse.

This paper is organized into six sections. Section 2 describes the basic working principle and nonlinear modeling of the current transformer. Section 3 compares existing insulation methods and describes an optimized insulation design process. Section 4 presents the experimental validation for the theoretical analysis. Section 5 provides a description of the results and their implications compared with the commercial current sensors and the state-of-the-art research work. Finally, Section 6 presents the conclusion of the study.

## 2. Working Principle and Nonlinear Modeling

Current transformer sensors are primarily used to reduce or “step-down” current levels. The high-current signal to be measured is transduced to an isolated output voltage that is compatible with the requirements of monitoring and control. The basic structure of a current transformer sensor is shown in Figure 2a. It typically consists of three parts: the secondary winding with *n* turns, a toroidal core, and a burden resistor *R_b_*. According to Ampere’s Law, an ideal equivalent circuit of a current transformer can be given as Figure 2b. The magnetic coupling between the primary conductor and the *n*-turns secondary winding is represented by the coupling coefficient *k*, which is ideally 1. When the time varying current *I_P_* flows through the center of the current transformer, it will generate an alternating magnetic field *B* that is almost entirely concentrated inside of the magnetic core. Then, a scaled-down current *I_Sec_*, 1/*n* of *I_P_*, is induced from the magnetic field *B* and generates the corresponding output voltage vs. on *R_b_*. Therefore, the output voltage vs. is proportional to the primary current *I_P_* and can be measured to accurately collect the current information of the power circuit. However, there are several factors that impact the accuracy of the current transformer in the field applications. To illustrate this point, a non-ideal lumped circuit is presented in Figure 2c. First, the coupling coefficient *k* is less than 1 because the primary side and secondary side are not fully coupled in practice, thus a leakage inductance *L_l_* is introduced. Second, parasitic elements exist in circuit and components including parasitic winding resistance *R_s_*, parasitic stray capacitance *C*, and equivalent series inductance (ESL) of *R_b_*. Furthermore, the effect of electromagnetic induction and the core losses can be equivalent to an extra branch of the introduced magnetizing inductance *L_m_* and resistance *R_c_*, leading to the current diversion *I_e_* from *I_Sec_*.

### 2.1. Accuracy Issues

In power electronics applications, switching current measurement is critical and has a high requirement on accuracy, which is taken as the scenario to illustrate the accuracy issues of the current transformer. Figure 3 shows two accuracy issues caused by the abovementioned factors: droop phenomenon in Figure 3a and bandwidth limitation in Figure 3b.

For droop phenomenon, the expression of *I_s_* that finally goes through the burden resistor *R_b_* is derived as (3). When a pulse current with a duration from several microseconds to milliseconds is measured, the current remains constant on such a large time scale. Accordingly, the current *I_C_* going through *C* can be ignored because the parasitic capacitance *C* is in picofarads and has a tera ohm impedance. Thus, the current error is mainly caused by the exciting current *I_e_*, which is the sum of the magnetizing current *I_m_* on *L_m_* and core loss current *I_R_* on *R_c_.* The magnetizing inductance *L_m_* can be calculated as (4).
(3)$$I_{s}=I_{S\mathrm{ec}}-I_{e}-I_{C}\approx I_{S\mathrm{ec}}-(I_{m}+I_{R})$$
(4)$$L_{m}=\frac{\mu_{0}\mu_{r}n^{2}A_{e}}{L_{e}}$$
where *μ*_0_ is the permeability of free space; *μ_r_* is the relative permeability, which is the ratio of the permeability of a specific material to the permeability of free space *μ*_0_; *A_e_* is the cross-section area of the toroidal core; and *L_e_* is the effective length of the toroidal core. In the frequency range of interest, the impedance of *R_c_* is often much larger than the magnetizing impedance, thus it can be neglected in the first analysis. Therefore, *L_m_* dominates the exciting current *I_e_*. When the voltage across *L_m_* during the on-time is assumed to be constant (which it is not, but close enough to simplify), *I_e_* increases with time as a ramp. This increasing current multiplied by the terminal resistance creates an increasing voltage drop, reducing the accuracy, as shown in Figure 3a. Consequently, the magnetizing inductance *L_m_* is critical to optimize the droop during the design.

Besides the droop phenomenon during the static state, the accuracy during the switching transient is also crucial. Generally, the pulse current is generated by switching power devices in power electronic applications. When wide bandgap devices are used, the switching transient can be tens of nanoseconds, which requires a high-frequency (30–50 MHz) acquisition. As shown in Figure 3b, the red waveform is the actual current to be sensed, while the blue waveform is the current measured by a current sensor with insufficient bandwidth. A smaller slew rate switching transit, smaller oscillation amplitude, and sometimes even severe distortion appear. All of these measurement errors are detrimental to some conditions such as current control or overcurrent protection and switching loss calculation.

To investigate how these factors affect the bandwidth, the transfer function for a high frequency is required. At a high frequency, the capacitive reactance of *C* is very small, which cannot be ignored, and *R_s_* is ignorable because the inductive reactance of *L_l_* and *L_b_* is far larger. Particularly noteworthy is the domination in the magnetizing branch of *R_c_* and *L_m_*. In conventional analysis, the resistance *R_c_* will dominate because the reactance of the magnetizing inductance is way too high at a high frequency, thus *L_m_* is ignored. However, those analyses do not include the degradation of *L_m_* at a high frequency. Generally, the permeability value of the magnetic core at a high frequency can drop to 0.1% of the value at a low frequency, as shown in Figure 4a. Therefore, the accuracy issue is mainly caused by *L_m_*, *C*, *L_l_*, and *L_b_.* The high-frequency equivalent circuit is simplified as Figure 4b. The transfer function for a high frequency is derived as (5):(5)$$G_{H}(s)=\frac{L_{m}R_{b}s}{CL_{b}\left(L_{l}+L_{m}\right)s^{3}+CR_{b}\left(L_{l}+L_{m}\right)s^{2}+\left(L_{b}+L_{l}+L_{m}\right)s+R_{b}}$$

As shown in the Bode plots in Figure 5, the lower limit frequency is inversely proportional to the value of the magnetizing inductance *L_m_*, while the higher limit frequency is also negatively correlated to the value of the stray capacitance *C* and ESL of *R_b_*. The impact of leakage inductance *L_l_* on the bandwidth is insignificant, but large enough leakage inductance reduces the gain of the system. Therefore, to design a high-bandwidth current transformer, a large *L_m_* and small *C*, *L_b_*, and *L_l_* are required.

In most of the research, *μ_r_* is regarded as a constant value, thus *L_m_* does not change over the entire frequency range. However, the *μ_r_* attenuation effect with frequency significantly influences *L_m_*. To accurately investigate the dramatic attenuation effect on the bandwidth droop phenomenon, a comprehensive modeling for nonlinear *L_m_* at a high frequency is necessary for guiding the current transformer design.

### 2.2. Nonlinear Modeling Analysis

Nonlinear feature of magnetizing branch.

The conventional linear modeling analysis has an assumption that the impedance of the magnetizing inductance becomes relatively high because *Z_m_* = *jwL_m_* at a high frequency. Therefore, *L_m_* is ignored and *R_c_* dominates the magnetizing branch in conventional high-frequency analysis. However, as shown in Figure 4a, the permeability of the magnetic core will drop dramatically in the high-frequency region, as mentioned. The permeability droops from 12,000 to 100 around 1 MHz for material 3E12, which causes the *L_m_* to become a very small value from (4), even at a high frequency.

Nonlinear behavioral modeling of *L_m_*.

To describe the nonlinear behavior of *L_m_*, a curve fitting for the permeability *μ_r_* is first required. As the permeability value varies in a large range on a wide frequency domain, the logarithmic scale for both *μ_r_* and frequency *f* are applied to perform the curve fitting. However, the commonly used models for curve fitting, such as linear, power functions, rational, sinusoid, polynomial, and exponential, either do not perform well or rely on a high-order expression to obtain a reasonable fit. The high-order function brings large complexity to the current transformer modeling. To address this issue, an arctangent-based fitting algorithm is proposed. Its general formula can be expressed as (6):f(x) = *a*_1_ × arctan(−*b*_1_ × x + *c*_1_) + *a*_2_ × arctan(−*b*_2_ × x + *c*_2_) + *a*_3_ × arctan(−*b*_3_ × x + *c*_3_) + *d*(6)

By optimizing the fitting with different parameter iterations, the comparison between several commonly used models and the proposed fitting result is shown in Figure 6a. It can be seen that the arctangent-based curve fitting matches the one from the datasheet with good accuracy. The optimized parameters used for the arctangent-based curve fitting are *a*_1_ = 0.6394, *b*_1_ = −1.783, *c*_1_ = −1.555, *a*_2_ = −3.704, *b*_2_ = 1.274, *c*_2_ = −4.4541, *a*_3_ = 5.682, *b*_3_ = 1.62, *c*_3_ = −0.174, and *d* = 1.982. Based on (4) and (6), the nonlinearity of the magnetizing inductance dependence on frequency *L_m_*(*f*) is plotted in Figure 6b. The most critical part to fit is the degradation range. The value of *L_m_*(*f*) based on the curve fitting and datasheet match well over that frequency range. Therefore, the fitting algorithm is good enough to demonstrate accuracy.

High-frequency response of a current transformer.

With the nonlinear magnetizing inductance *L_m_*(*f*), the transfer function can be further described as (7):(7)$$G_{H}(s)=\frac{L_{m}(f)R_{b}s}{CL_{b}\left(L_{l}+L_{m}(f)\right)s^{3}+CR_{b}\left(L_{l}+L_{m}(f)\right)s^{2}+\left(L_{b}+L_{l}+L_{m}(f)\right)s+R_{b}}$$

The magnitude of *G_H_*(*s*) is calculated as in (8):(8)$$Gain=20\cdot \log_{10}\sqrt{G_{H}(s)\cdot G_{H}{}^{*}(s)}$$
where *G_H_^*^*(*s*) is the conjugate of *G_H_*(*s*). Then, the frequency response with linearized *L_m_* and nonlinear *L_m_*(*f*) is as shown in Figure 7. When *L_m_* changes linearly from 30 μH to 1 mH as the dashed line, it only affects the low cut-off frequency. However, with the nonlinear *L_m_*(*f*), there is an obvious drop in the gain at a high frequency when all other parameters remain the same. To investigate this drop, the number of turns *n* varies from 5 to 20. It is worth noting that the gain drop becomes smaller with the increasing *n*. When *n* is larger than 10, the gain drop is compensated and almost consistent with the linearized *L_m_*. In addition, with the number of turns increasing, the stray capacitance increases, thus the higher bandwidth decreases. Therefore, there is a compromise between the gain drop and bandwidth. These results conclude that a too low or too high number of turns on the secondary side will both have a bandwidth limit at a high frequency. In summary, nonlinear modeling can more accurately describe the behavior of the current transformer and gives a more comprehensive description of the bandwidth.

### 2.3. Saturation Analysis

Besides the gain drop and bandwidth error that can be compensated for with optimized design, saturation is another intractable problem to which to pay attention. The current transformer is in saturation when the entire magnetic domain is aligned in the magnetic core, causing no more flux to increase even when the primary current changes. According to Faraday’s Law $v=N\frac{d\Phi }{dt}$ and Ampere’s Law NI=Hl, the magnetic to electrical relationship is as shown in Figure 8.

As shown in Figure 8b, the flux change is governed by the voltage applied to the windings, and the magnetic domain or the flux cannot change instantaneously because energy is required. To saturate the core, a specific magnetizing current *I_m_* is required, and the time to complete the flux change is a function of the voltage applied to windings. Combining the magnetizing current, voltage at the windings, and time, the energy loss of the core is the area between the electrical characteristic and the vertical axis. Therefore, a designed current transformer must, at the very minimum, be capable of sustaining the full internal flux built up until the moment of turn-off within a switching period. In addition, the magnetizing current should be fully reset within the turn-off period so that the full range of flux can be available for the next turn-on period. Based on the B–H curve of the magnetic core, Faraday’s Law, and volt-second balancing, to avoid saturation, two conditions must be satisfied:(9)$$\left\{\begin{array}{l} V_{\mu s(on)\_\max}=V_{on}t_{on}\leq N\cdot B_{\max}\cdot A_{e}\\ V_{\mu s(off)\_\max}=V_{off}t_{off}\geq V_{\mu s(on)\_\max} \end{array}\right.$$
where *B_max_* is the maximum flux density that the current transformer can handle before saturation; *V_on_* and *V_off_* are the voltage applied on the magnetizing inductance during the turn-on and turn-off switching period, respectively; and *t_on_* and *t_off_* are the switching on and off times, respectively. To analyze the saturation, switching current is explained as in Figure 9. The maximum voltage–second product can be determined by (10).
(10)$$\left\{\begin{array}{ll} V_{\mu s(on)\_\max} & =t_{on}\times (V_{L_{l}}+V_{R_{s}}+V_{L_{b}}+V_{R_{b}})\approx L_{l}\frac{dI_{s}}{d\tau_{r}}\tau_{r}+I_{s}\cdot R_{b}\cdot \tau_{on}\\ & {=I}_{s}(L_{l}+R_{b}\cdot \tau_{on})\\ V_{\mu s(off)\_\max} & =t_{off}\times V_{Lm}\approx L_{m}\frac{dI_{s}}{d\tau_{f}}\tau_{f} \end{array}\right.$$
where *τ_r_* and *τ_f_* are the current rising and falling time, respectively, and *τ_on_* is the stable current time. All the factors are maximized to account for the worst-case transient conditions. The contribution of *V_Rb_* varies directly with the line current. *V_Lb_* can be neglected because *L_b_* will be as small as possible to increase the accuracy. *V_RS_* is the smallest contributor and is neglected because of *R_s_* < *R_b_*. *V_Ll_* is developed by the *di*/*dt* of the sensed current and is not observable externally. However, its impact is considerable, given the sub-microsecond rise-time of the current signal.

Ideally, the current transformer can realize self-reset on some occasions because *L_m_* >> *L_l_* and, normally, turn-off transient is faster than turn-on transient, thus *V_us(off)_max_* > *V_us(on)_max_* can be guaranteed to avoid saturation. However, this is not always the case, especially with the widely varying duty cycles in circuit operations. It is worth noting that sustained unbalance in the on or off volt-second products leads to core saturation and a total loss of the current-sense signal. To avoid saturation, an external reset network, usually using a Zener diode to increase *V_μs(off)_*, is applied when necessary. The schematic of the circuit and switching waveforms is presented in Figure 10. There is one diode *D_1_* in series with the burden resistor *R_b_*. All of the waveforms are shown in Figure 10b. During the turn-off period, *D*_1_ will have the reverse recovery stage that generates a high peak voltage to reset the magnetizing current in a short amount of time. Then, the maximum voltage–second product is expressed in (11).
(11)$$\left\{\begin{array}{ll} V_{\mu s(on)\_\max} & =t_{on}\times (V_{L_{l}}+V_{R_{s}}+V_{L_{b}}+V_{R_{b}}+V_{F})\approx L_{l}\frac{dI_{s}}{d\tau_{r}}\tau_{r}+(I_{s}\cdot R_{b}+V_{F})\cdot \tau_{on}\\ & {=I}_{s}(L_{l}+R_{b}\cdot \tau_{on})+V_{F}\cdot \tau_{on}\\ V_{\mu s(off)\_\max} & =t_{off}\times V_{Lm}\approx V_{BR}\cdot t_{rr} \end{array}\right.$$
where *V_F_* is the forward voltage of the diode, *V_BR_* is the reverse recovery voltage, and *t_rr_* is the reverse recovery time. Commonly, *V_BR_* will be a large value, which will reset the circuit dramatically. Then, the saturation current transferred to the primary side can be derived by (12).
(12)$$\left\{\begin{array}{l} I_{P\_\max}=\frac{N^{2}B_{\max}A_{e}}{L_{l}+R_{b}\tau_{on}},\;\mathrm{without}\;\mathrm{external}\;\mathrm{reset}\;\mathrm{network}\\ I_{P\_\max}=\frac{N\cdot (NB_{\max}A_{e}-V_{F}\tau_{on})}{L_{l}+R_{b}\tau_{on}},\;\mathrm{with}\;\mathrm{external}\;\mathrm{reset}\;\mathrm{network}\; \end{array}\right.$$
assuming *I_P_* = *NI_s_*.

## 3. High-Voltage Insulation Design

With the increasing demand and research trend on high-voltage high-power-density power electronics systems, a higher insulation demand for the control or measurement side becomes a challenge. When more and more high-frequency and high-voltage wide bandgap devices spring up, all of the insulators of the current sensor will experience a high slew rate (*dv*/*dt*) (ranging from tens to hundreds of kV/*μs*) and repetitive voltage pulses (frequency ranging from hundreds of kHz to MHz), resulting in a fast and continuous insulation degradation [51]. Therefore, it is necessary to find a better insulator or better insulation method to guarantee good insulation and lifetime without changing the parasitic elements of the current sensor.

### 3.1. Comparison of Insulation Methods

As shown in Figure 11, there are three main methods for electrical and electronic insulation on current transformers: (1) enameled wires or rubber insulated wire, (2) Kapton tape, and (3) encapsulation.

Enameled wires have a very thin layer of varnish on the surface of the copper to enable the insulation, thus leading to a higher copper density in a coil. However, it is a surface coating, which is easily subject to cracking and abrasion. Rubber insulated wire by itself is not very commonly used as an insulation method. Because rubber tends to lose its characteristics with time, it will become rigid and break down easily under tension, especially for outdoor applications. However, it has more flexibility to set up in the system.

Kapton tape is mostly used as insulation across an electrical field, including capacitor insulation, transformer insulation, and wire insulation. It is known for its excellent insulating property, extreme adhesion, and wide temperature range compatibility from −20 °C to 350 °C. Furthermore, the minimum thickness can be as low as 0.03 mm. No other material can provide such excellent performance in such a small space. On top of that, it can be removed without any leftover residue. However, it has very poor resistance to mechanical wear, mainly abrasion within cable harnesses. Therefore, there is a high risk of loss of insulation during movement. In addition, it is difficult to apply it on some small gaps of the current transformer.

Encapsulant is widely used in electrical and electronic applications, especially embedded delicate electronic circuitry to produce void-free insulation around components. It offers higher performance, increased miniaturization, and long-term reliability for insulation. There are a variety of potting materials on the market and the two most commonly used types are silicone gel and epoxy resin. Silicone gel is suitable for conventional case-type package. Thanks to its good viscosity, it is allowed to flow around the component and fill any voids. After curing, the soft status of the material provides insulation with a minimum amount of stress. Epoxy resin has two sealing technologies: transfer molding (TM) and direct potting (DP). The TM encapsulated approach was initially investigated because of its environmental endurance and mechanical robustness. However, as the TM process is complicated and less flexible, direct potting (DP) epoxy resin is developed to perform as the traditional silicone gel, but at the same time provides good shock and vibration capability.

To provide a more clear and concise comparison, Table 1 summarizes the performance of these three methods based on four critical categories: electrical insulation, mechanical robustness, temperature withstand, and flexibility. The performance of each method is evaluated and ranked based on their respective index values, such as dielectric strength for electrical insulation, hardness for mechanical robustness, operation temperature for temperature withstand, and the ability to fill small gaps for flexibility. The rank of each method is denoted by the number of ★.

### 3.2. Parasitic Effects of Different Insulation Methods

While operating at a high frequency, more attention should be given to minimizing the parasitic elements such as leakage inductance and stray capacitance, as they will cause high peaks and severe oscillation and dramatically limit the bandwidth. Leakage inductance is mainly affected by the magnetic core geometry and winding arrangement, while stray capacitance is mainly affected by the winding arrangement and insulation type. Therefore, here, for typical current transformer design, the insulation methods’ influence on the stray capacitance is mainly analyzed.

As shown in Figure 12a, the stray capacitance includes three parts: turn-to-turn capacitance *C_tt_*, turn-to-core capacitance *C_tc_*, and winding-to-winding capacitance *C_ww_*. The turn-to-turn capacitance can also be divided into capacitance between adjacent turns *C_tt_* (*i*, *i* + 1) and capacitance between nonadjacent turns *C_tt_* (*i*, *i* + …). According to the winding geometry, capacitance between nonadjacent turns is smaller than the capacitance between adjacent turns. To simplify the analysis, capacitance between nonadjacent turns is neglected. Moreover, the distance between the primary winding to the secondary winding is usually large enough so that winding-to-winding capacitance can be neglected as well. The sketch of the coil is shown in Figure 12b. Assuming all the turns are uniformly wound and neglecting the turn curvature, the equivalent capacitance *C_eq_* can be described from [58] as (13).
(13)$$C_{eq}=\frac{\pi \cdot \varepsilon_{r}\cdot \varepsilon_{0}}{\ln({\left(1+t/r\right)}^{\varepsilon_{0}}\cdot {\left(B+\sqrt{B^{2}-1}\right)}^{\varepsilon_{r}})}$$
where
(14)$$B=\frac{p/2r}{\left(1+t/r\right)}$$
where *r* is the coil radius, *ε*_0_ is the permittivity of free space, *t* is the thickness of the insulation coating, *ε_r_* is the permittivity of the insulation material, and *p* is the distance between two adjacent turns.

The calculation results are summarized in Table 2. The magnetic core size is 22 mm (outer diameter), 14 mm (inner diameter), and 6.4 mm (height), assuming the number of turns is 10 and coils are uniformly wound on the core. Comparing the capacitance, epoxy resin potting has the lowest capacitance value.

Gathering all of the information from Table 1 and Table 2, epoxy resin was selected to serve as the insulation of the current transformer.

### 3.3. Optimization of the Insulation Process

One of the practical problems for encapsulation is how to encapsulate the current transformer to guarantee there is no air trapped inside. To solve this issue, an optimized encapsulation process is designed as in Figure 13. Firstly, a 3D-printed glass-filled nylon bobbin is applied as a container for the epoxy resin as well as the first layer of insulation. Second, epoxy resin is poured into the bobbin. Then, the current transformer is merged into the epoxy resin. It is worth noting that this process is not the usual procedure of pouring the gel into electronics devices. The reverse order can further guarantee less air trapped inside, and the current transformer is fully covered by the epoxy resin. Lastly, to ensure the removal of all of the air inside the encapsulation epoxy resin, degassing is executed in a degassing chamber, and then it is cured at 125 °C for four hours.

To evaluate the insulation ability of the encapsulated current transformer, a Hi-Pot test was carried out. Figure 14a presents the Hi-pot test setup: the high voltage side is applied from 500 V to 5 kV by reading the leakage current *I_l_* from Valhalla Scientific 5880 A Dielectric Analyzer, which is the primary cable with silicone rubber insulation; the low voltage side (ground) is connected to the terminal of the secondary side winding. Figure 14b presents the leakage current curve under different voltages. It is verified that this design can achieve at least 5 kV insulation with a 2 nA discharge current.

## 4. Experimental Evaluation

### 4.1. Design Process of the Current Transformer

The process of designing a current transformer typically involves several steps, as shown in Figure 15, including specifications of the system, magnetic core selection and modeling, and winding configuration. To design a current transformer, the application requirements first need to be identified, such as the primary current rating, maximum duty cycle, frequency range of operation, and output voltage range of the current transformer. The purpose of this paper is to design a current transformer specifically intended for measuring switching currents up 30 A and compatible with standard digital control systems. To ensure compatibility with an analog to digital converter that has a maximum input voltage of 3.3 V, it is necessary for the current transformer’s output voltage to be below this value. In power electronics applications, SiC devices are commonly utilized within a frequency range of 10 kHz to 100 kHz. When selecting the appropriate magnetic core material, it is important to consider several factors, including permeability, frequency response, and saturation. To achieve a higher inductance (*L_m_*), it is generally advantageous to choose a core material with higher permeability. The nonlinear modeling is built via the proposed curving based on the magnetic material. Next, there is a tradeoff between the number of turns and burden resistor. Once the maximum primary current and output voltage are determined, the ratio between the burden resistor and number of turns *R_b_*/*n* is fixed. A higher number of turns results in a higher burden resistance value, which means a smaller voltage drop and a more significant amplitude drop, and vice versa. Finally, a prototype of the current transformer is built and tested to verify if it meets the specifications and accuracy requirements. For power electronics applications, a compact design and saturation are important factors. Based on the drain to source current of SiC MSOFET measurement scenarios with a typical To-237 package, a magnetic core with a suitable size, that is, outer diameter of 22 mm, inner diameter of 14 mm, and height of 6 mm, is selected.

### 4.2. Droop Phenomenon Evaluation

To investigate the droop phenomenon, a half-bridge circuit is built to generate the continuous switching current. The schematic and test bench are shown in Figure 16. The power devices are 1.2 kV/19 A SiC MOSFETs (C2M0160120D) from Wolfspeed in Fayetteville, AR, USA. An isolated gate driver for the SiC MOSFET is designed with the recommended gate drive voltage range of 20/−5 V. A fixed gate resistance (*R_g_*) of 5 Ω is used. The 50 kHz PWM signal with 0.5 duty cycle for the high-side switch (S_1_) and low-side switch (S_2_) is complementary and generated by a microcontroller (Texas Instruments TMS320F28335). The current transformer is placed between the load and the negative terminal of the power supply. Another commercial 30 MHz Rogowski coil, Tektronix TRCP0300, is directly installed near the current transformer. The performance of the proposed current transformer is evaluated by comparing the results obtained from these two approaches.

Figure 17a,b illustrate how the number of turns and core material, respectively, can impact the current droop. Figure 17a demonstrates that the current transformer with 10 and 20 turns has good matching with the commercial Rogowski coil, while the current transformer with 5 turns indicates a droop of 1 A. Figure 17b compares current transformers with material 3E12 and 3C94, where *n* = 10. Material 3C94 has a permeability *μ_r_* = 2300, while material 3E12 has a higher permeability *μ_r_* = 12,000. A 0.8 A droop is observed on the output of the 3C94 current transformer, but the 3E12 current transformer performs well. Therefore, to minimize the current droop, it is recommended to use a larger number of turns and higher permeability magnetic core with the appropriate burden resistor and core size.

### 4.3. Saturation Phenomenon

In power electronics applications, especially those with high frequencies, it is crucial to reset the core completely before the next cycle begins. As mentioned earlier, using a diode in series with the burden resistor can help reset the core and reduce saturation. Figure 18 shows the comparison of the experimental results obtained with and without a diode in series, using the current transformer with 3E12 material and 10 turns. As shown in Figure 18a, the test result of the current transformer with a diode matches well with the 100 MHz DC current probe TCPA300, and no saturation is observed. As illustrated in Figure 18b, without a diode, the current transformer experiences a current drift as it can only measure AC. Additionally, during the turn-off period, the current transformer cannot be fully reset, resulting in its saturation.

To ensure that the magnetic core is fully reset during high-frequency operation, diode selection is also important. Figure 19 presents the gate driver signal *v_gs_* and a comparison of three types of diodes in series with the same current transformer design: ultra-fast (*t_rr_* < ~50 ns), fast (*t_rr_* < ~200 ns), and slow (*t_rr_* > 200 ns). Table 3 lists the parameters of the three types of diodes. The ultra-fast and fast diodes are both able to reset the core in a timely manner, whereas the slow diode is too slow and leads to core saturation. Furthermore, even though the ultra-fast diode resets the core faster, it creates larger voltages with harsher edges that will contribute to EMI noise. Therefore, the fast diode is preferred. It is also important to note that the diode is not only used for core reset, but also to protect the controller from the negative reset pulse.

### 4.4. Bandwidth Evaluation

Based on the droop evaluation and diode selection, ultimately, the material 3E12, a number of turns of 10, and a fast diode RS3AB-13-F are selected. To evaluate the bandwidth of the proposed current transformer, a double pulse test (DPT) setup is built to generate current pulses with very short rise and fall times. The DPT schematic and experimental setup is shown in Figure 20a,b, respectively. The setup typically includes a pulse generator, an isolated gate driver, a device under test (DUT), a freewheeling diode, and an air core inductor. The pulse generator is used to generate the multi-pulse waveform using the Texas Instruments TMS320F28335 DSP. The DUT is a 1.2 kV/63 A SiC MOSFET (C2M0025120D) from CREE. The upper freewheeling diode is a 1.2 kV/54 A SiC Schottky diode (C4D40120D). The isolated gate driver is designed with the recommended gate drive voltage range of 20/−5 V and an external gate resistance of 2.5 Ω. The air core inductor is connected in parallel with the upper freewheeling diode to provide a path for the fast-switching transient. The proposed current transformer and 30 MHz commercial Rogowski coil Tektronix TRCP0300 are placed between the DUT and the negative terminal of the power supply. By comparing the waveforms of the pulses passed through the DUT obtained by the proposed current transformer and the commercial current probes, the bandwidth of the proposed current transformer can be evaluated.

Figure 21 shows the measured waveforms from the multi-pulse experiment, including the gate driver signal *v_gs_* and drain current *I_D_* with 40 A load current. In Figure 21a, it can be observed that the output of the proposed current transformer matches well with the output of the commercial Rogowski coil. To further compare the details and high-frequency response, the turn-on and turn-off transient is shown in Figure 21b. It can be seen that the 74 ns turn-on and 34.8 ns turn-off transients and tens of megahertz oscillations are well-matched between the two measurements. Therefore, the high-frequency response of the proposed current transformer allows it to capture fast transients, such as the 34.8 ns turn-off transient, which would require a current sensor with a bandwidth of at least 30 to 50 MHz.

To further validate the bandwidth of the current transformer, a 100 MHz commercial DC current probe TCPA 300 is compared to a 30 MHz commercial Rogowski coil and the proposed current transformer in Figure 22. The switching transient has a 10 to 90% rise-time of 27 ns and the oscillation frequency is 21.28 MHz. It can be observed that the output of the proposed current transformer has a better high-frequency response than the 30 MHz commercial Rogowski coil and is comparable to the 100 MHz commercial DC current probe. As previously mentioned, it is recommended that the bandwidth of a current transformer should be at least three to five times higher than the highest frequency component of the signal being measured. In this case, because the highest frequency component is around 20 MHz, the bandwidth of the proposed current transformer should be from 60 to 100 MHz to accurately capture the rise time and oscillation. Moreover, in Figure 22, the waveform of the proposed current transformer matches really well with that from the commercial current probe; therefore, the bandwidth of the current transformer is close to 100 MHz.

### 4.5. Error Evaluation

The accuracy of the proposed current transformer, also called linearity error or non-linearity, is further proven in Figure 23. The linearity error is normally calculated as ((*v_s_* − *v_c_*)/*v_c_*) × 100%. In Figure 23a, it is shown that the experimental results (*v_s_*) match well with the calculated results (*v_c_*) in terms of linearity. It is worth noting that, at low currents, the linearity is better and, with the current increasing, the error also increases. In Figure 23b, the maximum error between the experimental results and calculation is 4% in the 50 A current range, which is accurate enough for current control and protection. This suggests that the accuracy of the current transformer may decrease at higher current levels. It is important to consider the maximum current range required for the application and design a current transformer with appropriate accuracy in that range.

## 5. Discussion

To provide a clear overview of the current transformer design, the comparison among the commercial products, state-of-the-art research work, and the proposed current transformer is shown in Table 4. The proposed current transformer performs better than the selected commercial current sensors in terms of cost and size. Furthermore, the commercial current sensor solutions with higher bandwidth target laboratory equipment rather than real-time control applications. For the state-of-art research work, a cascaded connection using two stages of current sensor is popular, but it would not fundamentally solve the size issue and complicates the measurement even further. Especially, when the commercial current sensor is implemented in the second stage, the cost of the measurement solution increases significantly. The proposed current transformer has a sufficient bandwidth for most power electronic applications and a relatively small size. Additionally, the proposed current transformer has excellent performance in terms of cost optimization.

## 6. Conclusions

This paper presents a comprehensive design guide of a high-frequency current transformer for switching current measurement in power electronics applications. Compared with the conventional design process, a nonlinear modeling of the magnetizing inductance across a wide frequency range is proposed to increase the accuracy of the bandwidth estimation. In addition, several considerations in the current transformer design such as droop phenomenon, saturation issues, and insulation design are discussed in detail. Finally, the proposed current transformer is validated under 50 kHz continuous 10 μs current pulses. Future work will focus more on minimizing the size to integrate it within power modules.

## Figures and Tables

**Figure 1 sensors-23-04626-f001:**
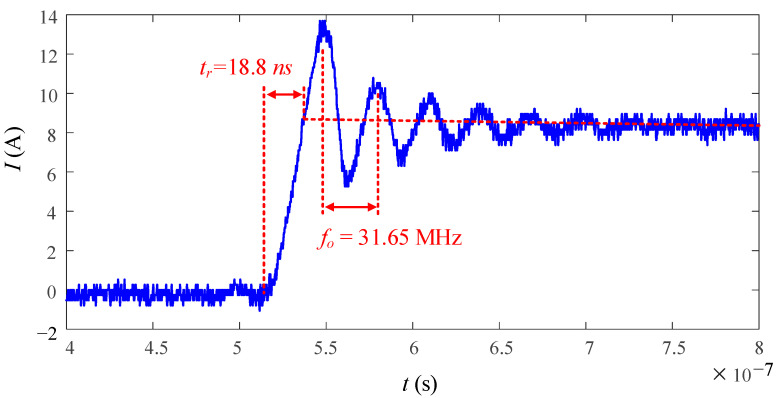
An example of a switching transient of WBG devices.

**Figure 2 sensors-23-04626-f002:**
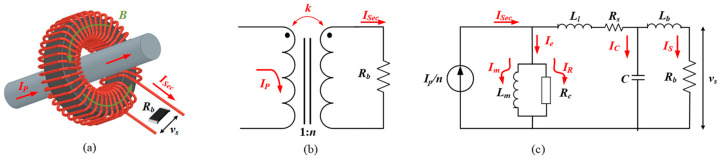
Construction and modeling of a current transformer: (**a**) construction sketch of a current transformer; (**b**) ideal equivalent circuit; (**c**) lumped equivalent circuit with all parasitic parameters.

**Figure 3 sensors-23-04626-f003:**
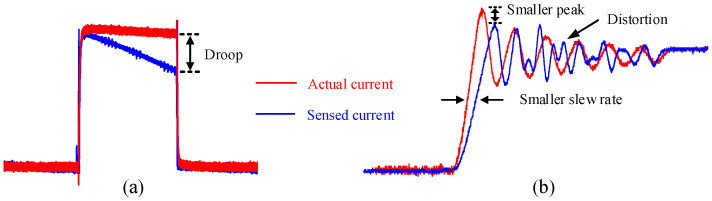
Accuracy issues: (**a**) droop phenomenon caused by increasing exciting current; (**b**) bandwidth limitation.

**Figure 4 sensors-23-04626-f004:**
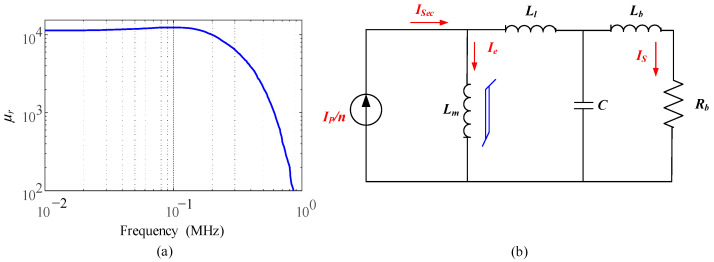
High-frequency modeling: (**a**) nonlinear characteristic of *μ_r_*; (**b**) simplified equivalent circuit at a high frequency.

**Figure 5 sensors-23-04626-f005:**
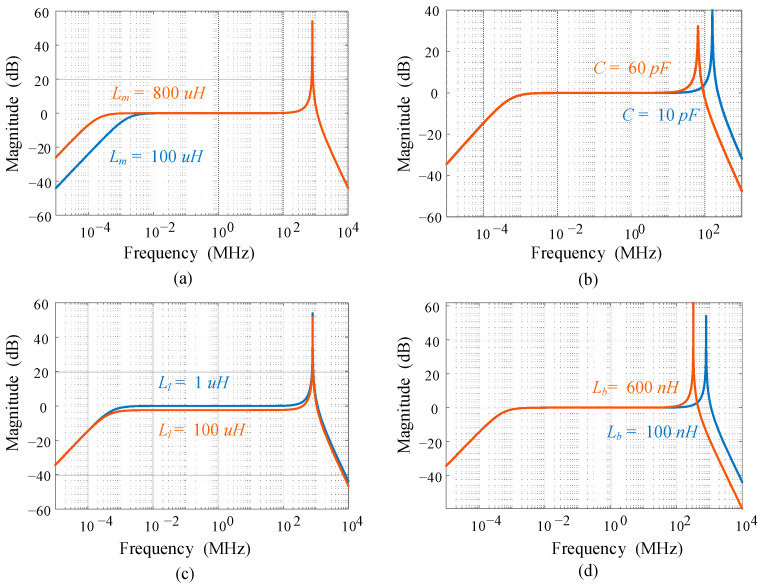
Bode plots with different factors: (**a**) magnetizing inductance *L_m_*; (**b**) stray capacitance *C*; (**c**) leakage inductance *L_l_*; (**d**) ESL of *R_b_*.

**Figure 6 sensors-23-04626-f006:**
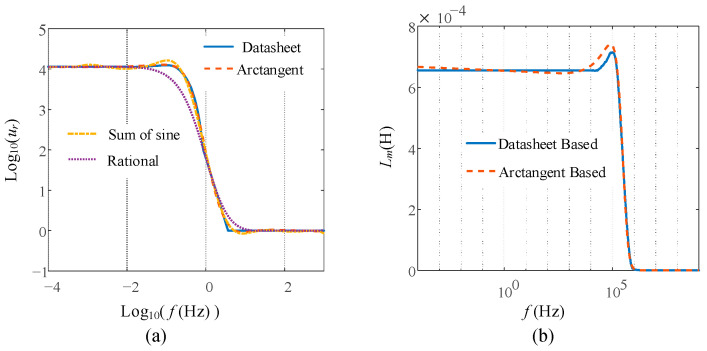
Nonlinear modeling: (**a**) curve fitting based on different models; (**b**) comparison *L_m_*(*f*) between datasheet-based and curve-fitting-based modeling.

**Figure 7 sensors-23-04626-f007:**
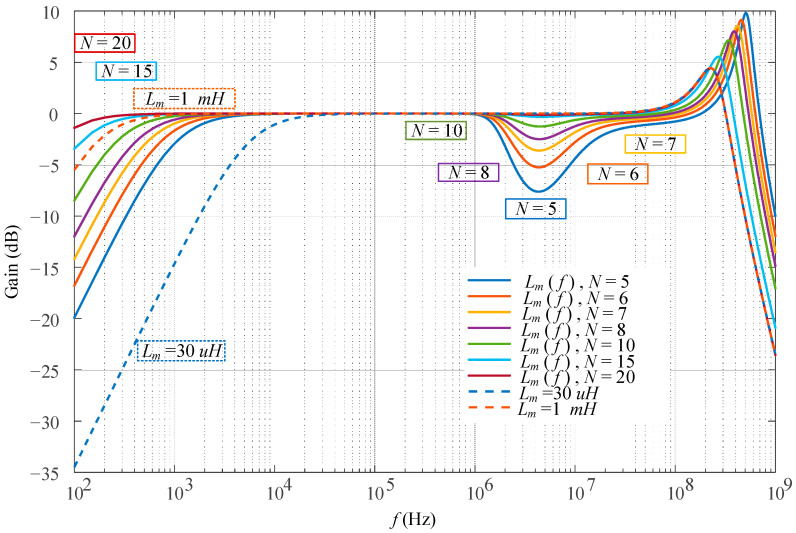
Gain response at a high frequency with nonlinear *L_m_*.

**Figure 8 sensors-23-04626-f008:**
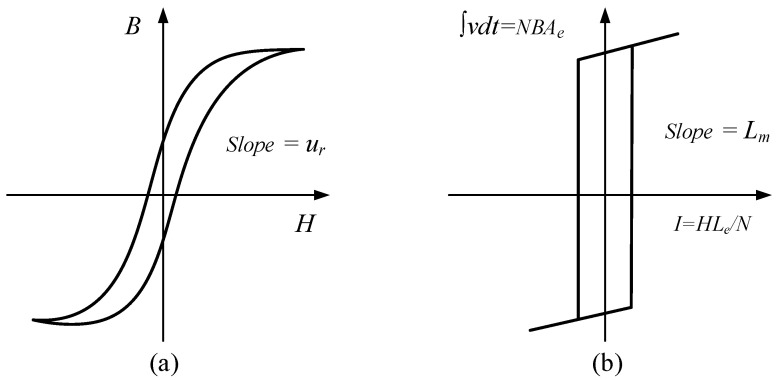
Saturation characteristic: (**a**) magnetic core B–H curve; (**b**) equivalent electrical characteristic.

**Figure 9 sensors-23-04626-f009:**
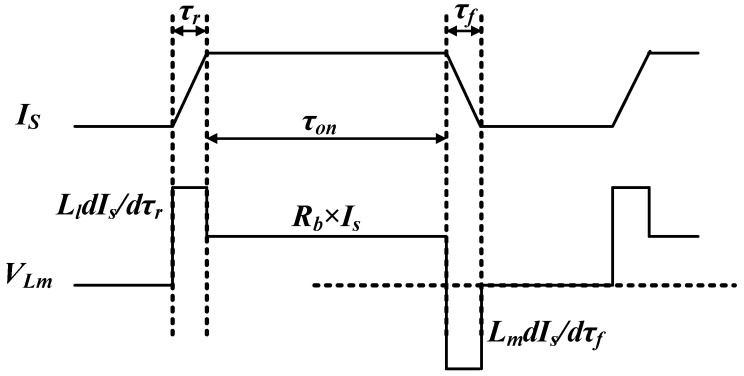
Switching waveforms of the secondary side.

**Figure 10 sensors-23-04626-f010:**
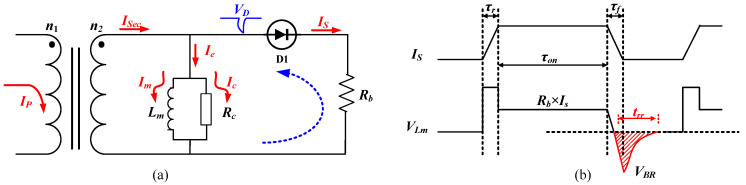
Current transformer with external reset circuit: (**a**) schematic; (**b**) switching waveforms of the secondary side.

**Figure 11 sensors-23-04626-f011:**
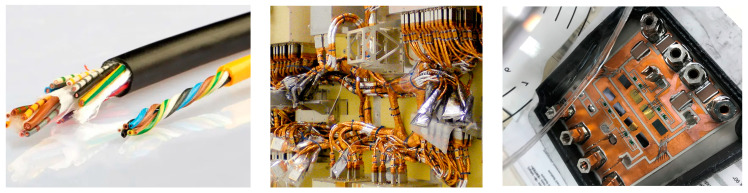
Common insulation methods for current transformers [52,53,54].

**Figure 12 sensors-23-04626-f012:**
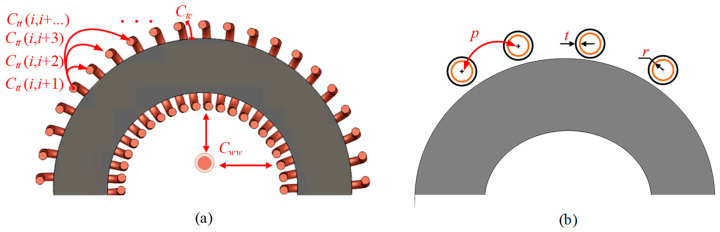
Stray capacitance: (**a**) cross-section view; (**b**) sketch.

**Figure 13 sensors-23-04626-f013:**
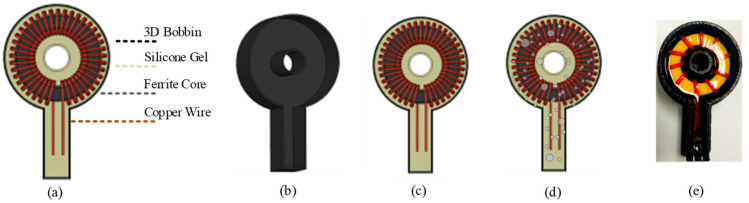
Optimized insulation process: (**a**) overall review; (**b**) 3D bobbin; (**c**) encapsulation; (**d**) degassing; (**e**) profile display.

**Figure 14 sensors-23-04626-f014:**
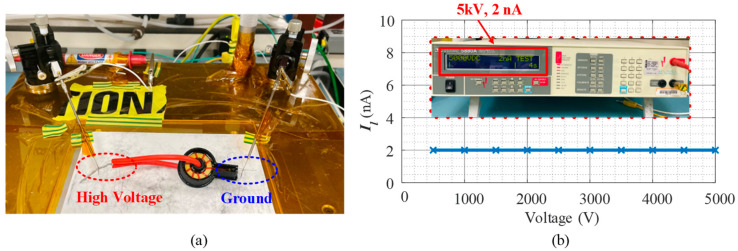
Hi-pot test of the encapsulated current transformer: (**a**) setup; (**b**) leakage current.

**Figure 15 sensors-23-04626-f015:**
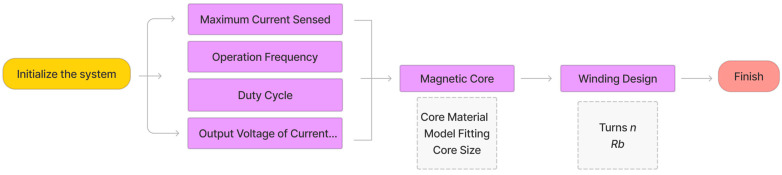
Design process for the current transformer sensor.

**Figure 16 sensors-23-04626-f016:**
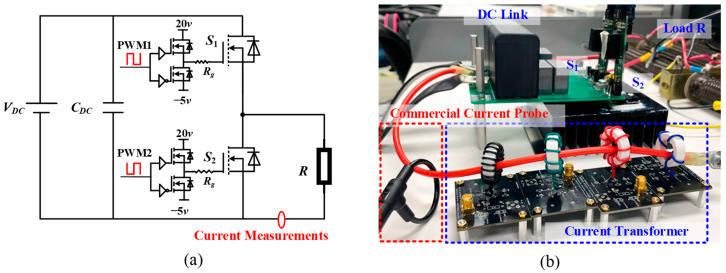
Half-bridge circuit: (**a**) schematic; (**b**) experimental setup.

**Figure 17 sensors-23-04626-f017:**
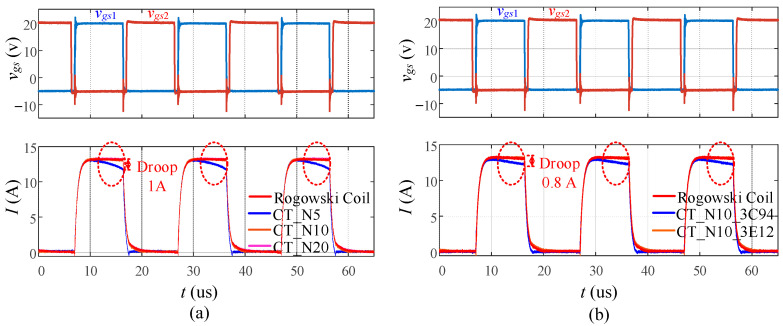
Droop phenomenon with (**a**) different turn ratios *n*; (**b**) different materials.

**Figure 18 sensors-23-04626-f018:**
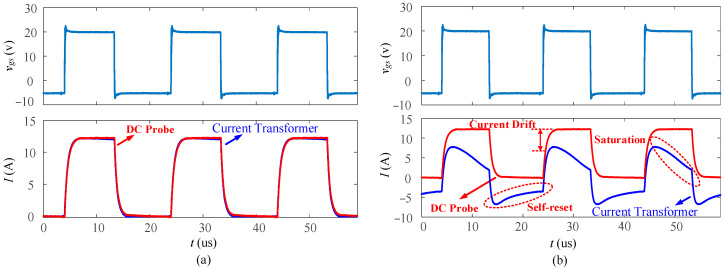
Saturation phenomenon: (**a**) with a diode; (**b**) without a diode.

**Figure 19 sensors-23-04626-f019:**
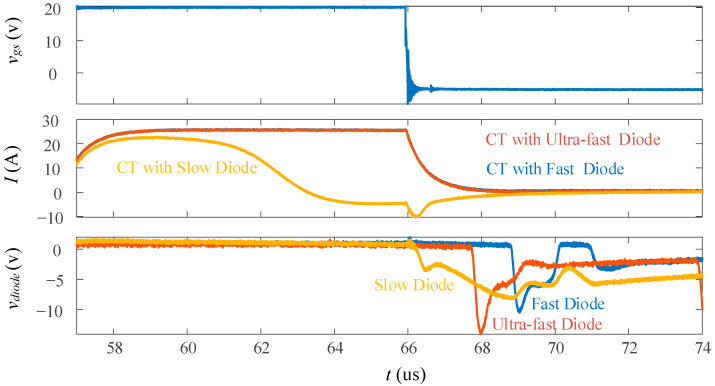
Reset response using various diodes: ultra-fast, fast, and slow.

**Figure 20 sensors-23-04626-f020:**
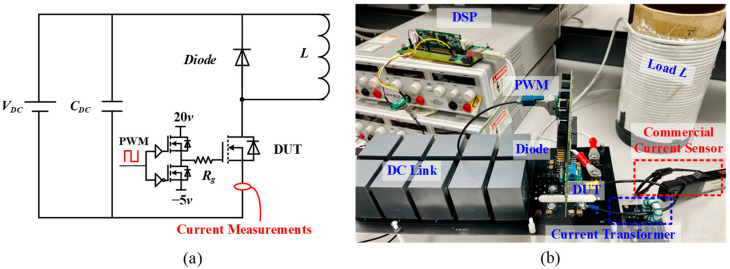
DPT test bench: (**a**) schematic; (**b**) experimental setup.

**Figure 21 sensors-23-04626-f021:**
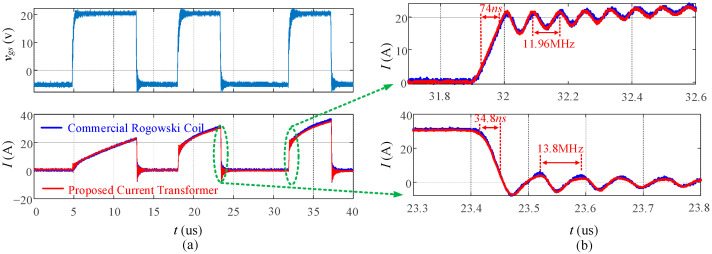
Multi-pulse measurement: (**a**) overall review; (**b**) turn-on and turn-off transit.

**Figure 22 sensors-23-04626-f022:**
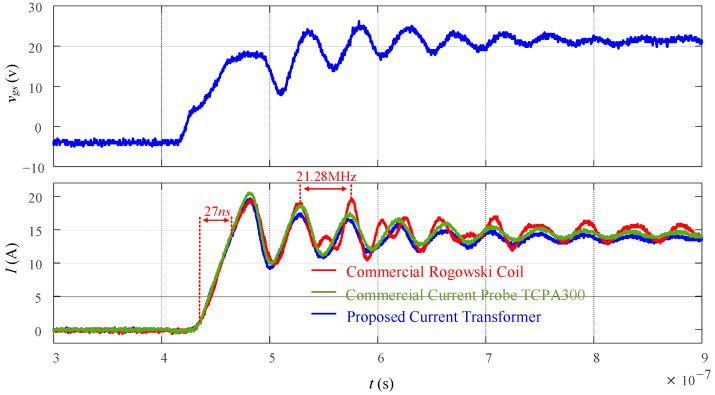
Bandwidth validation waveforms.

**Figure 23 sensors-23-04626-f023:**
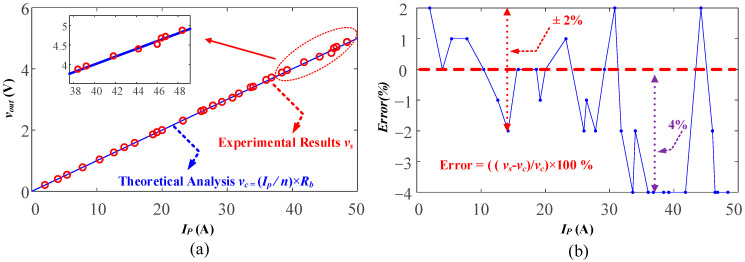
Error evaluations: (**a**) linearity; (**b**) error.

**Table 1 sensors-23-04626-t001:** Comparison among the three common insulation methods.

	Enameled Wires or Rubber Insulated Wire	Kapton Tape [53]	Encapsulation	
			Epoxy Resin [55]	Silicone Gel [56]
Electrical Insulation Dielectric Strength (v/mil)	★ ^1^ (200–500)	★★★ (7k)	★★ (500)	★★ (500)
Mechanical Robustness Hardness (Shore) [57]	★★ (Shore 00–60)	★★ -	★★★ (Shore D-80)	★ (Shore 00–35)
Temperature Withstand Operation Temp. (°C)	★★ (240/105)	★★★ (−20 to 350)	★★★ (−50 to 315)	★★ (−50 to 210)
Flexibility (Small-gap filling)	★	★★	★★★	★★★

^1^ A higher number of stars ★ indicates superior performance within a given category.

**Table 2 sensors-23-04626-t002:** Turn-to-turn capacitance for different insulation methods.

	Enameled Wires or Rubber Insulated Wire	Kapton Tape	Encapsulation	
			Epoxy Resin	Silicone Gel
*r*	0.4 mm	0.4 mm	0.4 mm	0.4 mm
*t*	0.5 mm	1 mm	5.4 mm	5.4 mm
*ε* _0_	8.854 × 10^−12^ F/m			
*ε_r_*	3	3.3	4.15	2.9
*p*	5.4 mm			
*C_eq_*	1.37 × 10^−11^	1.68 × 10^−11^	1.12 × 10^−11^	1.26 × 10^−11^

**Table 3 sensors-23-04626-t003:** Parameters for the three types of diodes: ultra-fast, fast, and slow.

Diode Type	Reverse Recovery Time (*t_rr_*)	Peak Reverse Voltage (*V_r_*)	Forward Current (*I_f_*)	Product Number
Ultra-fast	25 ns	50 V	3 A	ES3AB-13-F
Fast	150 ns	50 V	3 A	RS3AB-13-F
Slow	2.5 μs	50 V	3 A	S3A-E3/57T

**Table 4 sensors-23-04626-t004:** Comparison among the existing current sensors.

Reference	Bandwidth	Estimated Cost	Insulation	Size (mm)
Commercial products				
Hall effect sensor [43]	1 MHz	$10	Yes	/
Coaxial current shunt [44]	2 GHz	$290	No	/
Rogowski coil [46]	50 MHz	$1800	2 kV	100 mm circumference coil
Current transformer [47]	250 MHz	$800	Yes	34.8 × 20 × 9
Research work in literature				
Current transformer cascaded with current probe [31]	60 MHz	$5200	Yes	200 (L) × 16 (W) × 32 (H)
Current transformer cascaded with commercial current transformer [13]	/	$700	Yes	38.6 × 12.7 × 16.3
Two current transformers cascaded [49]	70 MHz	/	Yes	27 × 20 × 17
Proposed current transformer				
/	100 MHz	$20	>5 kV	22 × 14 × 6

## Data Availability

Not applicable.

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
