# Peer review of "A Nonlinear-Model-Based High-Bandwidth Current Sensor Design for Switching Current Measurement of Wide Bandgap Devices"

_sensors, 2023, doi:10.3390/s23104626_

Round 1

Reviewer 1 Report

In the abstract, to briefly describe how the authors simulate for non-linear inductance.

Is Figure 1 of the authors' own work to claim the typical switching transient? It needs to put citation.

The challenges and recommendation in line 46-96 seem to be of general information, which is not suitable for Introduction. The authors should include more publications, and detail out the research gap/motivation of the study. Further, for some points, no ref is cited. And in some statements/points, only one ref was cited. Introduction really need a serious improvement.

The statement that the bandwidth must be 5x higher, needs to cite the relevant ref.

The use of even though in line 119 does not seem suitable.

Figure 5, on Y-axis, to correct the spelling magnitude.

In line 297, is 'combing' the right word?

How did the authors come up with excellent, poor etc in Table 1?

K in Figure 14 should be a small k

In line 594, is it correct to say that sensitivity of the CT dependent on the current levels? Hence different test load been used to verify this?

Reviewer 2 Report

A few minor formatting issues were noticed, such as if you notice the caption of figure 8.  

Author Response

  • Thank you very much for taking the time to review our paper and providing your valuable feedback. We have carefully reviewed the entire paper and made the necessary changes to address the formatting issues in the revised version of this paper. Specifically, the caption of Figure 8 has been updated to ensure consistency and clarity.
  • We greatly appreciate your attention to detail and your help in improving the quality of our paper. Thank you again for your time and effort in reviewing our paper.

Reviewer 3 Report

The author describes “A High-Bandwidth, Low-Cost Current Sensor for Power Electronics Applications”.

Ø  Lack of clarity in the article title, it should provide the contribution and novelty of the article with a clear application.

Ø  The paper is well described and can be accepted for the publication

Author Response

Thank you very much for taking the time to review our article. The authors have responded to the comments and listed the answers in the following parts.

Q1: Ø Lack of clarity in the article title, it should provide the contribution and novelty of the article with a clear application.

  • Authors fully agree with your suggestion that the current title could be improved, and the title of the article has been revised to “A Nonlinear Model based High-Bandwidth Current Sensor design for Switching Current Measurement of Wide Bandgap Devices,” to better convey the contribution and novelty of our work, along with a clearer application.
  • The main contribution and novelty of this article is the nonlinear modeling of the magnetizing inductance for the current transformer based on the arctangent-based fitting algorithm. In conventional analysis, the magnetizing inductance is assumed to be fixed, which can result in inaccurate bandwidth estimation and suboptimal design process. With the comprehensive analysis for the optimization design from the bandwidth, droop error and core saturation point of view, a high bandwidth of 100 MHz and high insulation capability up to 5 kV current transformer design for measuring the switching current of wide bandgap devices is proposed. The application of the current sensor “switching current measurement of wide bandgap devices” has been added to the title. We sincerely appreciate your valuable feedback, and we hope that our revisions have met your expectations and improved the clarity of our article.

 Q2: Ø The paper is well described and can be accepted for the publication

  •  We sincerely appreciate your positive feedback on the quality of our work and are delighted to hear that you found it well-described. We will continue to strive for excellence in our research and thank you for your time and expertise in reviewing our work once again.

Reviewer 4 Report

the draft in present form is ready to be published without any errors found. 

Author Response

Thank you very much for your time and effort in reviewing our paper. We are thrilled to receive your positive feedback on the quality of our work and appreciate your kind words. We have carefully reviewed the entire paper and made the necessary changes to address any errors that may have existed. Thank you once again for your time and expertise in reviewing our work.